# Molecular Mechanisms of Senescence and Implications for the Treatment of Myeloid Malignancies

**DOI:** 10.3390/cancers13040612

**Published:** 2021-02-04

**Authors:** Philipp Ernst, Florian H. Heidel

**Affiliations:** 1Internal Medicine 2, Hematology and Oncology, Jena University Hospital, 07747 Jena, Germany; Philipp.Ernst@med.uni-jena.de; 2Research Program “Else Kröner-Forschungskolleg AntiAge“, Jena University Hospital, 07747 Jena, Germany; 3Internal Medicine C, Hematology and Oncology, Stem Cell Transplantation and Palliative Care, Greifswald University Medicine, 17475 Greifswald, Germany; 4Leibniz Institute on Aging, Fritz-Lipmann Institute, 07745 Jena, Germany

**Keywords:** senescence, SASP, leukemia, myeloproliferative neoplasms, myelodysplastic syndrome, senolytics

## Abstract

**Simple Summary:**

Senescence is a cellular program in response to stress and can prevent the expansion of pre-malignant and malignant cells by cell cycle arrest. However, long-term pro-tumorigenic effects have been described. This review aims to compile molecular mechanisms of cellular senescence and to discuss potential therapeutic implications in myeloid malignancies.

**Abstract:**

Senescence is a cellular state that is involved in aging-associated diseases but may also prohibit the development of pre-cancerous lesions and tumor growth. Senescent cells are actively secreting chemo- and cytokines, and this senescence-associated secretory phenotype (SASP) can contribute to both early anti-tumorigenic and long-term pro-tumorigenic effects. Recently, complex mechanisms of cellular senescence and their influence on cellular processes have been defined in more detail and, therefore, facilitate translational development of targeted therapies. In this review, we aim to discuss major molecular pathways involved in cellular senescence and potential therapeutic strategies, with a specific focus on myeloid malignancies.

## 1. Introduction on Cellular Senescence

The inability of somatic mammalian cells to undergo further replication was first described as cellular senescence in 1961 [1]. This physiologically limited replicative capacity, known as the “Hayflick limit”, results from successive shortening of telomeres during each cell division [2,3]. Telomerase, a ribonucleoprotein enzyme complex that counteracts telomere shortening, is transcriptionally silent in normal human somatic cells, except in certain stem cells, including hematopoietic stem cells, but can be reactivated in neoplastic cells [2,4,5]. Senescent cells are unable to replicate because of halted DNA synthesis due to an arrested cell cycle in the G_1_/S interphase [6,7]. In addition, these cells are not able to respond to physiological mitogenic stimuli, although the number of receptors on the surface remains unchanged [8]. In contrast to senescent cells, quiescent or pre-senescent cells are nonreplicating, diploid cells in the G_0_ phase that are able to restore proliferation after suitable stimuli. The only senescent cells that can re-enter the cell cycle under certain conditions are senescent tumor cells [9]. In addition, replicative senescence state cells can develop so-called premature senescence independent of their biologic or in vitro culture age. This state can be induced by oncogenes, including Ras and factors that cause DNA damage, such as oxidative stress, alcohol, and bacterial lipopolysaccharides [10,11,12,13]. Increased expression of anti-apoptotic B-cell lymphoma 2 (BCL-2) protein and cyclin-dependent kinase (CDK) inhibitors p16^INK4a^ and p21^WAF1^, as well as reduced proapoptotic caspase activity, are considered typical features of senescent cells [14,15,16,17,18]. Furthermore, the formation of a senescence-associated secretory phenotype (SASP) is characteristic, which among growth factors, proteases, and other factors is mainly characterized by the secretion of pro-inflammatory cytokines and chemokines. In addition to autocrine senescence maintenance, these factors can exert opposing effects on the surrounding microenvironment with both senescence-inducing and mitogenic stimulation [19,20].

A widely used method for histochemical labeling of senescent but not pre-senescent or immortalized cells in vitro and in vivo is the senescence-associated β-galactosidase (SA β-gal) assay [21]. β-Galactosidase is a ubiquitous lysosomal hydrolase, and their detection in senescent cells takes place at pH 6. The validity of this method alone is controversial because an unspecific expression of SA β-gal could also be demonstrated in immortalized tumor cell lines [22]. However, to support other senescence indicators, such as phosphorylated histone H2AX (γH2AX) or trimethylated lysine 9 of histone 3 (H3K9me3) leading to positive senescence-associated heterochromatin foci (SAHF) staining, the SA ß-gal assay is still considered a useful marker [23]. Although cellular senescence is a possible cellular mechanism to avoid malignant transformation, it may also promote cancer development by altering the cellular microenvironment through the SASP factors [24].

## 2. Molecular Mechanisms of Senescence

### 2.1. DNA Damage Response (DDR)

DNA damage response (DDR) is an important mechanism for the initiation of the senescence pathway. In the context of replicative senescence, progressive shortening of telomeres leads to telomere uncapping, which is sensed as double-stranded DNA breaks and thereby activates DDR [25]. On the other hand, DDR can be caused by oncogene-induced DNA replication stress, leading to telomere dysfunction, which seems to be a critical tumor-suppressive mechanism [26]. Ataxia telangiectasia mutated (ATM) and ATM-Rad3 related (ATR) are phosphatidylinositol 3-kinases (PI3K) and important components in repairing DNA damage and maintaining telomere length [27]. ATM and ATR bind to damaged DNA sites and phosphorylate histone H2AX. While ATM is primarily activated by double-stranded DNA breaks (DSB), ATR binds in association with ATR-interacting protein (ATRIP) to Replication protein A (RPA)-labeled single-stranded DNA (ssDNA) created by replication arrests [28,29]. By local changes in chromatin structure, γH2AX leads to recruiting the DNA repair complex with a focal assembly of the checkpoint kinases CHK1 and CHK2, which can then be phosphorylated by ATR and ATM. Both ATM and phosphorylated CHK2 at Threonine 68 are able to activate p53 by phosphorylation, which leads to the induction of cell cycle arrest [30]. While the ATM-CHK2 axis plays an auxiliary role, especially in response to DSB, the ATR-CHK1 axis is the main effector of DNA damage and replication control points [31].

### 2.2. Senescence Associated Molecular Pathways Are Interconnected

Depending on the trigger, different pathways are activated to induce senescence. As already described, p53 is activated by a variety of triggers, such as DDR, critical telomere shortening, or activation of oncogenic signaling pathways (e.g., RAS-signaling). The activity of p53 is mainly controlled by the E3-ubiquitin ligase Mouse double minute 2 homolog (MDM2). The transcription of *MDM2* is, in turn, induced by p53 as a negative feedback loop since MDM2 acts as a negative regulator of p53 by direct binding and degradation via the ubiquitin–proteasome complex [32,33]. MDMX (also known as MDM4), another endogenous inhibitor of p53, promotes MDM2 activation and inhibits p53 transactivation [34]. When activated in response to oncogenic RAS, the intranuclear protein p14 binds MDM2 in the nucleus and thereby prevents the degradation of p53 [35,36]. Likewise, in response to DSB, MDM2 is inhibited by phosphorylation through activated ATM [37]. Independent of DDR, the translation of p53 is enhanced by AKT by inducing the mammalian target of the rapamycin complex 1 (mTORC1) [35]. The PI3K/AKT pathway is in turn negatively regulated by the phosphatase PTEN [35,38]. The activity of p53 is also influenced by post-translational modifications. In response to oncogenic RAS, promyelocytic leukemia protein (PML) activates p53 through acetylation [39]. Similarly, coactivators p300 and CREB-binding protein (CBP) also lead to acetylation of p53 through stress stimuli, such as UV radiation, hypoxia, and alkylating agents [16]. The activity of p300 and CBP is in turn suppressed by MDM2 [40]. Silent information regulator two ortholog 1 (SIRT1) belongs to the family of sirtuins, a group of highly conserved proteins with NAD^+^-dependent protein deacetylase activity. By deacetylation of p53, SIRT1 counteracts PML-mediated senescence [41]. SIRT1 is, in turn, a target gene of miR-34c-5p and miR-181a, which inhibit its expression and thus reduce deacetylation of p53 [42,43,44]. The immediately activated target gene after activation of p53 is the p21^WAF1^ gene, which encodes the cyclin-dependent kinase 2 (CDK2) inhibitor p21^WAF1^. The inhibition of CDK2 by p21^WAF1^ leads to decreased phosphorylation of the retinoblastoma protein (RB).

In contrast to DDR and telomere dependent mechanisms, the ERK-p38^MAPK^ pathway is activated independently of the telomere function but typically in the presence of oxidative stress or oncogenic signaling. Oncogenic RAS induces premature senescence by sequential activation of MEK-extracellular signal-regulated kinase (ERK). Accumulated MAP kinase kinases, MKK3 and MKK6, activate p38^MAPK^ that thereby is able to degrade BMI1, a member of the Polycomb group gene family and transcriptional repressor of p16^INK4a^ [45]. In addition, the MAPK/ERK cascade stimulates the activity of ETS1, which causes an increased transcription of p16^INK4a^ [46]. Upregulated p16^INK4a^ forms complexes with the RB kinases CDK4 and CDK6, which leads to long-term inhibition and stabilization of the hypo-phosphorylated state of RB.

As previously discussed, the final link of both pathways is formed by RB (Figure 1). RB is a key regulator to initiate senescence, as its phosphorylation state can determine the progression of cells from the G1 to S phase of the cell cycle. In contrast, hypophosphorylated RB forms a complex with the transcription factor E2F. In addition to suppression of E2F activity, the RB/E2F complex leads to the recruitment of histone methyltransferases, which results in focal heterochromatin formation of E2F target gene promoters through transcriptional repressive trimethylation of H3K9 [47]. As a result, S-phase-promoting genes remain stably suppressed and cells remain in growth arrest in the late G1 phase.

### 2.3. The Senescence Associated Secretory Phenotype (SASP)

Senescent cells are usually able to express a set of molecules that define a so-called ‘Senescence Associated Secretory Phenotype’ (SASP). These factors include a variety of chemokines, pro-inflammatory cytokines, proteases, growth factors, and non-protein molecules (Figure 2). The phenotype of SASP may vary since transcriptional regulation of SASP-associated molecules is highly heterogeneous among different cell types [48]. Different mechanisms have been described that are responsible for initiating the expression of SASP factors. In the context of DDR, the accumulation of cytoplasmic chromatin fragments has been described that are released from the nuclei of primary senescent cells [49]. The release of genomic DNA from the nucleus into the cytoplasm of senescent cells is caused by autophagic degradation of lamin B1, the major structural component of the nuclear membrane [50]. Cyclic guanosine monophosphate (GMP)-adenosine monophosphate (AMP) synthase (cGAS) is a cytosolic DNA sensor that recognizes cytoplasmatic DNA in the form of cyclic dinucleotides found during pathogenic infections [51]. Cyclic GMP-AMP (cGAMP), which is synthesized by cGAS, in turn, activates the 379 amino acid protein stimulator of interferon genes (STING) located in the endoplasmic reticulum [52,53]. Activated STING forms a complex with TANK-binding kinase 1 (TBK1) and shifts to perinuclear regions, where TBK1, released to endolysosomal compartments, is able to phosphorylate the transcription factors interferon regulatory factor 3 (IRF3) and nuclear factor ‘kappa-light-chain-enhancer’ of activated B-cells (NF-κB) and thereby enables translocation to the nucleus [54,55].

Another activator of the SASP is the transcription factor GATA4 [56]. Under normal conditions, GATA4 is degraded through the autophagic adaptor protein p62 by selective autophagy. The mechanism by which GATA4 is recognized by p62 remains elusive so far. However, p62 can bind ubiquitinated proteins with its UBA domain [57]. In response to DNA damage, phosphorylation of the deubiquitinase USP28 by DDR kinases, ATM, and ATR leads to deubiquitination of either GATA4 or its associated factor [58]. Thus, GATA4 is stabilized and able to activate the SASP via NF-κB activation [56]. At the same time, overactivation of SASP is also prohibited. MiR-146a inhibits the expression of TNF receptor-associated factor 6 (TRAF6) and IL-1 receptor-associated kinase 1 (IRAK1) and, therefore, acts as a negative regulator of NF-κB [59]. In addition to NF-κB, which upregulates miR-146a expression as a negative feedback mechanism during inflammatory stimuli [60], expression of miR-146a is also stimulated by GATA4 [56].

Regulation of SASP has also been linked to JAK2/STAT3 signaling [61]. This pathway is activated in a specific subtype of p53-mediated senescence, initiated by an acute inactivation of PTEN [62]. In pancreatic tumor cells with loss of PTEN-induced senescence, the mainly immunosuppressive SASP could be reprogrammed by inhibiting the JAK2/STAT3 pathway, thus enabling tumor clearance by immunostimulatory cytokines [61]. P53 is required for the growth arrest in senescent cells but not for the induction of the SASP [63]. However, cells with non-functional p53 secrete significantly higher levels of most SASP factors. P53, therefore, restrains the SASP by negative feedback mechanisms to reduce the development of a pro-inflammatory tissue environment. One feedback loop is the negative regulation of USP28 by p53, thereby contributing to the attenuation of the SASP [58]. Other mechanisms include the restriction of p38^MAPK^ activation, a stress-inducible kinase that acts as a DDR-independent regulator of the SASP [64].

Importantly, NF-κB signaling represents a bottleneck in the development of the SASP. In addition to described upstream kinase cascades, the activation of the NF-κB system is usually connected with several pattern recognition receptor pathways, e.g., toll-like receptors (TLR) and inflammasomes [65]. The transcriptional capacity of the NF-kB factors can be regulated by various kinases, of which inhibitory κB (IκB) kinases (IKK) α/β and NF-κB inducing kinase (NIK) are the most important. The NF-κB transcription factors RelA/p65, c-Rel, RelB, p50/p105, and p52/p100 are located in the cytoplasm in a dimerized state and inhibited by binding to the IκB proteins IκBα, IκBβ, IκBγ, IκBδ, IκBε, IκBζ, and Bcl3 [66]. After IκB proteins have been phosphorylated by activating kinases, they are released from the complex to be degraded by proteasomes. The NF-κB transcription factors are then able to translocate into the nucleus and transactivate the expression of certain target genes that define the SASP [66].

Another pro-inflammatory transcription factor, to which most SASP regulators converge, is CCAAT/enhancer-binding protein β (C/EBPβ). In collaboration with NF-κB, C/EBPβ regulates many SASP components and inflammatory cytokines [19,56,67]. During premature senescence, there is transient upregulation of NOTCH homolog 1 intracellular domain (N1ICD), which inhibits C/EBPβ. Transforming growth factor β (TGFβ) is upregulated together with N1ICD and leads to the expression of an immunosuppressive secretory phenotype. A decrease in N1ICD over time disinhibits C/EBPβ, which together with NF-κB stimulates the expression of pro-inflammatory cytokines (e.g., IL1a, IL6, and IL8) [67]. IL-6, a pleiotropic cytokine and SASP factor, acts mitogenic on neighboring cells on the one hand and is required cell-autonomously, on the other hand, to reinforce senescence by activating the promotor of *CDKN2B* encoding p15^INK4b^ that promote p16^INK4a^ by suppressing CDK4 and CDK6 [19].

## 3. Treatment Strategies for Myeloid Malignancies that Involve Cellular Senescence

Myeloid malignancies are clonal disorders of the hematopoietic stem or progenitor cells that are caused by genetic mutations or epigenetic alterations. These mutations lead to changes in proliferation, differentiation, or self-renewal. Myeloid malignancies include chronic stages, such as myelodysplastic syndromes (MDS), myeloproliferative neoplasms (MPN), and acute stages, such as acute myeloid leukemia (AML). Recently, preclinical studies and early clinical trials have highlighted several molecular targets that are effective in the treatment of myeloid malignancies either by additional senescence induction or by eradication of senescent cells (Table 1).

### 3.1. CDK and PARP Inhibition

The specific CDK4/6 inhibitor palbociclib not only blocks the proliferation of cancer cells but also induces senescence by the concomitant occurrence of CDK inhibition and disruption of proteasomal homeostasis [71,72]. The latter is mediated by reduced proteasomal association of the adapter protein ECM29, which normally acts as a proteasome inhibitor [72]. Palbociclib is already approved by the U.S. Food and Drug Administration (FDA) for the treatment of advanced breast cancer. Myelotoxicity, especially neutropenia, is the main reason for dose reduction or withdrawal. The proliferation of AML cells can be inhibited in vitro by palbociclib in a dose-dependent manner [73]. As a transcription complex component, CDK6 is localized at the promoter of the fms, such as the tyrosine kinase 3 (FLT3) gene [74]. When combined with FLT3 inhibitors, treatment with palbociclib demonstrated a synergistic effect on FLT3-mutated AML cells [74,75,76].

ADP-ribosylation by poly (ADP-ribose) polymerase 1 (PARP1) is an important mechanism in DNA damage repair. In addition to inducing synthetic lethality associated with a deficit in homologous recombination, PARP inhibition leads to cellular senescence trigger by DDR after DNA damage accumulation [77]. The oral PARP inhibitor olaparib is established, especially for the treatment of advanced germline BRCA-mutated ovarian and breast cancer [78,79]. Preclinical data also show the efficacy of PARP inhibitors in various subtypes of AML, either as monotherapy or in combination with classical chemotherapy, despite the absence of driving genetic mutations affecting DNA repair genes in most cases [80]. Furthermore, the addition of PARP inhibition to decitabine demonstrated a synergistic effect on MDS cells, so the combination therapy is currently being investigated further in clinical trials [80,81].

### 3.2. Inhibition of MDM2/MDMX

MDM2 inhibition induces a senescence-like state and has been shown to attenuate SASP [82,83]. Up to 47% of AML patients show overexpression of MDM2, which, therefore, appears as a common cause of functional p53 inactivation in de novo AML [84]. Likewise, MDMX is overexpressed in up to 97% of AML cases and 52% of MDS cases [85]. While mutations in *TP53* are a frequent event in specific subtypes of solid tumors, they occur in less than 10% of AML patients and, if so, mostly associated with complex aberrant karyotype in therapy-related AML [86,87]. For this reason, MDM2 and MDMX have been in focus as potential senolytic targets in myeloid cancers and are developed as targeted therapies in AML and MPN [88,89]. ALRN-6924 is a stapled α-helical peptide that shows dual MDM2/MDMX inhibitory potential and has recently been tested as a therapeutic approach in AML [88]. By inducing cell cycle arrest and apoptosis, ALRN-6924 demonstrated antiproliferative potential in AML cell lines and primary AML patient cells with wild-type p53 and has already been tested on patients with AML and MDS in a phase I clinical trial [90]. The MDM2 inhibitor ATSP-7041 also binds at phosphorylated p300, which additionally reduces the transcriptional efficacy of p53 [91]. However, since MDM2 is one of the major negative regulators of p53, its inhibition also poses the risk of off-target effects and possibly even the eradication of non-senescent cells [92].

### 3.3. Inhibition of SIRT1

Sirtinol, a specific inhibitor of SIRT1, induces senescence through increasing acetylation of p53 [93]. Sirtinol induces premature senescence and growth arrest in CML cells and, therefore, may represent an alternative strategy for TKI resistance [94]. Regarding AML biology, an miR-34c-5p deficiency was found in leukemia stem cells (LSC) of AML patients when compared to normal hematopoietic stem cells (HSC) and has been associated with poor prognosis [42]. Conversely, an increased miR-34c-5p expression can promote the eradication of AML stem cells. RAB27b is a RAB-GTPase that regulates intracellular trafficking of multivesicular endosomes through its effector protein SLAC2B, leading to exosome secretion [95]. Through this mechanism, RAB27b regulates the exosomal clearance of miR-34c-5p [42,95]. Selective targeting of RAB27b or SLAC2b could, therefore, represent a promising strategy to inhibit miR-34c-5p-loaded exosome release and induce senescence in LSCs via increased inhibition of SIRT1.

In various cell types, including MLL-rearranged AML cells, SIRT1 was shown to be downregulated by miR-181a [43,44,96]. In patients with cytogenetically normal AML, high miR-181a levels are associated with a better prognosis [97]. MiR-181a expression is directly modulated by the C/EBPα-p30 isoform, whose expression can, in turn, be enhanced by lenalidomide, leading to increased sensitivity of leukemia cells to chemotherapy [98].

### 3.4. Inhibition of Telomerase Activity

Disrupted telomere homeostasis can result in genomic instability in hematopoietic cells, causing myeloid diseases, such as MDS and AML [98]. Increased telomerase activity and shortened telomere length are also risk factors for the progression or relapse in patients with AML [99]. In addition to stabilizing proteins, the telomerase complex is characterized by reverse transcriptase and an internal RNA template, which are encoded on the telomerase reverse transcriptase (TERT) and telomerase RNA component (TERC) genes, respectively. Imetelstat is an oligonucleotide that competitively binds to the template region of the RNA component of telomerase. The resulting inhibition of telomerase-mediated DNA repair processes leads to growth disadvantage and increased susceptibility to genotoxic agents in malignant hematopoietic cells [100]. Imetelstat has already been evaluated in Phase II trials in MPN patients and showed promising response rates in patients with essential thrombocythemia and primary myelofibrosis, although higher grade myelotoxicity was also evident [101,102,103]. Furthermore, treatment with imetelstat showed disease control with markedly decreasing transfusion frequency in a phase II/III study in patients with low-risk MDS [104].

### 3.5. Epigenetic Modifiers

The aberrant expression of the Polycomb complex member and epigenetic regulator BMI1 is associated with several human malignancies. BMI1 is highly expressed in specific AML subgroups and correlates with therapy failure [105,106]. Inhibition of Histone deacetylases (HDACs) can suppress transcription of BMI1 and induce cell cycle arrest, senescence, and cell death in tumor cells [107,108,109,110]. Chidamide, a novel benzamide-type HDAC inhibitor, has been reported to induce G1 arrest and induces the death of LSC [111]. In refractory or relapsed AML cells, chidamide increases sensitivity to anthracyclines synergistically with decitabine induces apoptosis [112,113]. Furthermore, the addition of chidamide to imatinib was effective in overcoming imatinib resistance in T315I-positive CML [114].

The trimethylation of H3K9 leads to stable repression of S-phase-relevant genes. Histone lysine-specific demethylase 4C (KDM4C) is a 2-oxoglutarate-dependent Jumonji C family member that demethylates trimethylated lysine residues of H1.4K26 and di- and trimethylated lysine residues of H3K9 and H3K36 [115]. KDM4C was found to be overexpressed in various cancer entities, including leukemia [115]. Inhibition of KDM4C successfully restored senescence and effectively controlled melanoma cell growth in vitro and in vivo [116]. Consistently, genetic knockout of Kdm4c in MLL-AF9 leukemia targets AML cells in vivo, due to decreased interleukin 3 receptor α (IL3rα) subunit expression without disturbing healthy hematopoiesis [117,118]. Using the KDM4C-specific inhibitor SD70 [119], leukemia development could be inhibited in both syngeneic mouse models and patient-derived xenografts [120]. In human leukemia cells, JAK2 phosphorylates the tyrosine residue of H3Y41 and thereby excludes the transcriptional repressive heterochromatin protein HP1α from the cMYC promotor [121]. In addition, the binding of HP1α to a cMYC promotor is also inhibited by demethylation of trimethylated lysine residues of H3K9 and H1.4K26 by KDM4C [122]. Since JAK2 and KDM4C are both localized on the chromosome band 9p24 and this region is frequently amplified in primary mediastinal B cell lymphoma and Hodgkin lymphoma [123], simultaneous inhibition of JAK2 and KDM4C may be synergistic in treating these diseases [122]. Whether other JAK2-mutated cancers, such as myeloproliferative neoplasms, are dependent on KDM4C activity and whether these JAK2-mutated cells are sensitive to combinatorial treatment, remains so far elusive.

### 3.6. Influencing the Bone Marrow Microenvironment

The tumor microenvironment is crucial for the progression of various solid and hematologic malignancies. Via NOX2-derived superoxide production, AML cells are able to induce p16^INK4a^ associated senescence in bone marrow stromal cells that promote the survival of AML cells by the secretion of SASP factors, especially IL-6 and MIP-3α [124]. Leukemia treatment with purine analogs and anthracyclines causes therapy-associated senescence in leukemia cells as well as in the healthy bone marrow microenvironment. Senescent cells have the ability to recruit Chemokine (C-C motif) ligand 2 receptor (CCR2) positive immature myeloid cells (iMC) via the secretion of the SASP factor CCL2. In a tumor-free cellular context, these iMCs can differentiate into dendritic cells, macrophages, or neutrophils, which can remove senescent cells along with T cells and NK cells, a mechanism called senescence surveillance [125]. However, in the presence of tumor cells, tumor-secreted factors may prevent the differentiation of iMCs. Since iMCs inhibit immune cells, such as T cells, macrophages, dendritic cells, and NK cells, they are able to create an immunotolerant environment that allows tumor cells to proliferate. Therefore, they are described as tumor-associated myeloid-derived suppressive cells (MDSCs) [126]. Here, a vicious cycle may develop where preserved senescent cells further recruit iMCs, that enhance the ability of tumor cells to escape the immune system [127,128]. Therapeutic approaches could include interruption of the CCL2/CCR2 axis. In an AML mouse model, it has already been shown that administration of the CCL2 Spiegelmer mNOX-E36 abrogated infiltration of immunosuppressive M2-like macrophages within the leukemia microenvironment [129]. However, blockade of the CCL2-CCR2 axis is challenging since the majority of AML blasts also express CCR2. Accordingly, a blockade of the CCL2-CCR2 axis in mice leads to the release of leukemia cells from the bone marrow into the peripheral blood, and the combination with purine analogs has no synergistic effects [130]. 

Chronic myeloid leukemia (CML) can be effectively treated with tyrosine kinase inhibitors (TKIs), such as imatinib. However, in rare cases, patients develop resistance to TKI therapy, which occurs independently of mutations in the BCR-ABL tyrosine kinase domain. Hyaluronan, a component of the bone marrow microenvironment, inhibits senescence induction of imatinib treated CML cell lines since survival signals in CML cells are triggered not only by BCR-ABL but also by hyaluronan through the activation of the PI3K/AKT pathway [131,132]. Besides its production in stromal bone marrow cells, hyaluronan is also synthesized by leukemic cells themselves [131]. 4-methylumbelliferone (4MU) is an inhibitor of hyaluronan synthesis by depleting cellular UDP-glucuronic acid and the downregulation of hyaluronan synthase 2 and 3 [133]. With the addition of 4MU to imatinib, a synergistic effect on senescence induction in CML cells has already been demonstrated in vitro, suggesting an improved therapeutic response by reducing hyaluronan accumulation [132,134].

### 3.7. Senolytics

The senescent microenvironment not only has a negative impact on leukemia development but also affects normal hematopoiesis, which may explain the development of resistance, disease recurrence, and long-term side effects of cytostatic drugs [124,135]. Therefore, therapeutic elimination of senescent cells from the bone marrow microenvironment may represent a valid therapeutic strategy. As part of the innate immune system, NK cells are able to recognize senescent cells and senescent tumor cells via their C-type lectin-like receptor NKG2D. A prerequisite for the recognition and elimination of senescent tumor cells is that they express NKG2D ligands, primarily MIC-A and ULBP2, whose upregulation is autocrine-enhanced by type 1 interferons [136]. However, tumor cells are able to evade the NKG2D response, and, moreover, NK cells can be inhibited by recruited iMCs [127]. In this scenario, senolytic drugs would be warranted, with the ability to deactivate senescence-associated anti-apoptotic signaling pathways and protect senescent cells from proapoptotic stimuli of the microenvironment.

Navitoclax (ABT-263) is the first orally bioavailable inhibitor of the BCL-2 protein family and binds with high affinity to the anti-apoptotic proteins BCL-XL, BCL-2, and BCL-W [137,138,139]. Recently, navitoclax was described as a potent senolytic drug in aged mice and irradiated younger mice since it selectively eradicates senescent cells [140]. As platelets require BCL-XL for survival [141], navitoclax is usually toxic by inducing thrombocytopenia [142,143,144]. The conversion of navitoclax to the BCL-XL proteolysis targeting chimera PZ15227 binds BCL-XL to the E3 ligase cereblon for degradation, significantly reducing toxicity to platelets due to their low cereblon expression [145]. While the efficacy of navitoclax has previously been studied in lymphoid and myeloproliferative neoplasms [139,146,147], venetoclax, a BCL-2 inhibitor with a favorable safety profile, has shown considerable efficacy in the treatment of AML [148,149]. In combination with azacitidine, venetoclax leads to the inhibition of energy metabolism in LSCs of elderly AML patients and shows promising clinical activity in a patient population with historically poor outcomes [150].

Expression and phosphorylation of focal adhesion kinase (FAK) at its SRC-dependent phosphorylation site Tyr577 was increased in senescent cells, thus showing a reduced apoptotic response to cytotoxic stress [151]. Furthermore, it has been shown that activated SRC kinase phosphorylates Y281 and Y302 of MDM2, resulting in an increase in MDM2 stability and that SRC causes inhibition of therapy-induced senescence via upregulation of cMYC, a negative regulator of p21^WAF1^ [152,153]. Unlike other TKIs, such as Imatinib, which are not senolytic, dasatinib promotes apoptosis of senescent cells caused, in part, by inhibiting SRC kinase [154]. Treatment with a combination of dasatinib and quercetin, a flavonoid present in many fruits and vegetables, decreases the senescent cell burden in humans and prevents physical dysfunction in mice by alleviating multiple age- and senescence-related conditions, such as osteoporosis, cardiac dysfunction, or diabetes [155,156,157,158,159]. However, in addition to its approval for the treatment of CML, dasatinib is also being investigated as a senolytic in other myeloid neoplasms. Dasatinib and navitoclax appeared strongly synergistic in AML cells driven by NUP98-NSD1 and FLT3-ITD oncogenes [160]. Furthermore, dasatinib has been identified to reverse microenvironment-induced resistance to FLT3 inhibitors in AML [161].

## 4. Conclusions

Considerable progress has been made in understanding the molecular regulatory mechanisms of senescence and its biological function in tissue aging and tumorigenesis, which appears to be highly dependent on cell type. Molecular specification of senescence depends on the type of stimulus as well as the environmental conditions of the cell [162]. Senescent cells are secretory active, making SASP a common but highly heterogeneous senescence-associated feature [66,68,69]. The majority of secreted molecules are associated with inflammation and have the potential to both benefit and harm the cell and its environment [69,135]. Many cytostatic drugs used for tumor treatment may induce therapy-induced senescence in tumor cells and their surrounding microenvironment [9,152,163]. So far, a rather limited spectrum of compounds with the ability to specifically activate senescence pathways in myeloid cancers has been described. Therefore, additional therapeutics need to be investigated for myeloid malignancies to avoid collateral senescence damage by conventional cytostatic drugs. Elimination of senescent cancer cells could contribute to tumor therapy and avoid long-term effects triggered by the SASP [164,165]. In preclinical studies, BCL-2 and SRC inhibitors have been shown to eradicate senescent myeloid tumor cells and have already been tested in AML [148,149,160]. The addition of senolytic drugs can be considered an important advance in the control of hematologic cancers and, therefore, needs to be further developed for the pan-myeloid spectrum of neoplasms, such as AML, MDS, or MPN [166,167,168]. Finally, the discovery that programmed cell death 1 ligand 2 (PD-L2) is upregulated on senescent tumor cells [169] may open novel avenues to develop immunomodulatory senolytics in the near future.

## Figures and Tables

**Figure 1 cancers-13-00612-f001:**
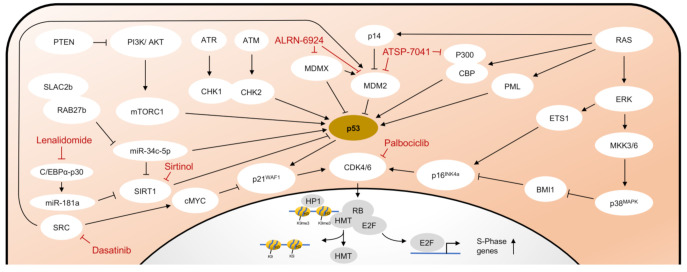
Pathways in cellular senescence. Schematic representation of the major pathways of cellular senescence and selected compounds for senescence induction or senolysis. HMT, Histone methyltransferase. Other abbreviations are explained in the text in detail.

**Figure 2 cancers-13-00612-f002:**
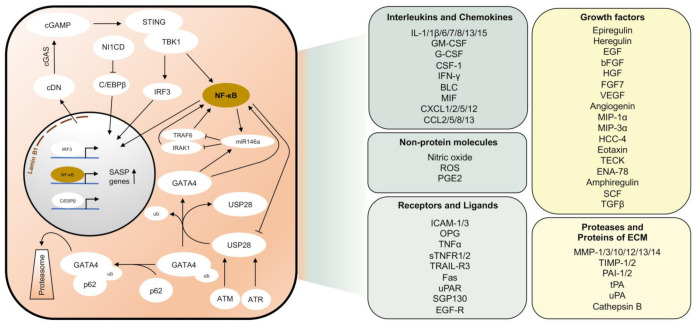
Pathways responsible for the induction of the senescence-associated secretory phenotype (SASP). Schematic representation of major pathways that are activated in senescent cells leading to the expression of SASP genes (left panel). Most important SASP factors according to [68,69,70] (right panel). cDN, cyclic dinucleotides. ECM, Extracellular matrix. ub, Ubiquitin.

**Table 1 cancers-13-00612-t001:** Selected active clinical studies investigating senescence-associated drugs on myeloid malignancies.

Drug	ClinicalTrials.gov ID	Phase	Target	Combined Drug	Diseases	Status	Estimated Study Completion Date
Palbociclib	NCT03132454	I	CDK4/6	Sorafenib or Decitabine or Dexamethasone or alone	Relapsed/refractory acute leukemia	Recruiting	June 2021
	NCT03844997	I/II	CDK4/6	CPX-351	Primary AML	Recruiting	January 2021
Talazoparib	NCT02878785	I/II	PARP	Decitabine	Primary AML, Relapsed/refractory AML	Active, not recruiting	December 2022
Veliparib	NCT03289910	II	PARP	Carboplatin or Topotecan	Primary AML, Relapsed/refractory AML, sAML arising from MDS, aCML, CMML, ET, PV, Myelofibrosis	Recruiting	June 2021
Imetelstat	NCT04576156	III	Telomerase	none	Intermediate-2 or high-risk Myelofibrosis	Recruiting	May 2024
Chidamide	NCT03031262	I/II	HDAC	Cytarabine	Postremission therapy of CBF-AML	Recruiting	December 2022
	NCT03453255	I/II	HDAC	Decitabine, Homoharringtonine, and Cytarabine	Postremission therapy of AML with t(8;21)	Unknown	December 2020
Navitoclax	NCT04472598	III	BCL-2, BCL-XL, BCL-W	Ruxolitinib	Myelofibrosis	Recruiting	July 2028
	NCT04041050	I	BCL-2, BCL-XL, BCL-W	Ruxolitinib or Celecoxib or alone	MPN	Recruiting	May 2023
Venetoclax	NCT03826992	I	BCL-2	CPX-351	Relapsed/refractory acute leukemia	Recruiting	January 2023
	NCT03573024	II	BCL-2	Azacitidine	Primary AML	Recruiting	June 2023
Dasatinib	NCT02013648	III	SRC	Cytarabine and Daunorubicin or Idarubicin	Primary CBF-AML	Recruiting	February 2024

CBF, Core-binding factor. aCML, atypical CML. CMML, chronic myelomonocytic leukemia. ET, essential thrombocythemia. PV, polycythemia vera.

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
