# Peer review of "Molecular Mechanisms of Senescence and Implications for the Treatment of Myeloid Malignancies"

_cancers, 2021, doi:10.3390/cancers13040612_

Round 1

Reviewer 1 Report

The authors prepared a nice review about an essential cellular process that determines cell fate and its implications for the treatment of myeloid malignancies. However, in paragraph 3 it should be added that further drugs are in clinical trials used for the treatment of myeloid malignancies. Suppression of telomerase reverse transcriptase (TERT) including TERT-targeting mRNA antisense oligonucleotides, short-interfering (si) RNA, peptides, and small molecules is studied in myeloid malignancies. Imetelstat is an oligonucleotide that targets and binds with high affinity to the active site of telomerase. Imetalstat was developed by Geron and is in a phase 2/3 clinical trial in lower-risk myelodysplastic syndromes (MDS). References: Ma W, Mason CN, Chen P et al. Telomerase inhibition with imetelstat eradicates β-catenin activated blast crisis chronic myeloid leukemia stem cells. Blood 2016; 128(22) 3065; Kishtagari A, Watts J. Biological and clinical implications of telomere dysfunction in myeloid malignancies. Ther Adv Hematol 2017; 8(11) 317-326; Hidaka D, Onozawa M, Miyashita N, et al. Imetelstat sensitizes hematopoietic malignancy cells to genotoxic agent via suppression of telomerase mediated DNA repair process. Blood 2019; 134(Supplement 1): 3369; Ma W, Balaian L, Mondala P, et al. Blood 2020; 136(Supplement 1): 18. 

Targeting ADP-ribosylation by poly(ADP-ribose polymerase (PARP) inhibitors is a further way to treat myeloid malignancies, mainly in combination with classical drugs. Olaparib (Lynparza) is an oral PARP inhibitor developed by Astra Zeneca was used in MDS and AML. References: Bochum S, Berger S, Martens UM. Olaparib. Recent Results Cancer Res 2018; 211: 217-233; Faraoni I, Giansanti M, Voso MT, et al. Targeting ADP-ribosylation by PARP inhibitors in acute myeloid leukaemia and related disorders. Biochem Pharmacol 2019; 167: 133-148; Faraoni I, Consalvo MI, Aloisio F, et al. Cytotoxicity and differentiating effect of the poly(ADP-ribose) polymerase inhibitor olaparib in myelodysplastic syndromes. Cancers 2019: 11: 1373 and several other articles.

Author Response

We thank reviewer 1 for the valuable comments and agree on the suggestions. Senescence induction by telomerase inhibition as well as PARP inhibition were thankfully received and added as a new chapter (in sections 3.1. and 3.4.).

Reviewer 2 Report

  1. Line35: hardly to not detectable
  2. Line55-57: A hallmark of senescent cells is the exhibition of a senescence-associated secretory phenotype (SASP) mainly consisting of pro-inflammatory cytokines and chemokines, which carries out a complex crosstalk both at the paracrine level and in the microenvironment [20, 21]. Could the author rephrase this sentence? A little bit confusing and incoherent.
  3. Line60-61: The reorganization of the cytoskeleton with an increase in actin stress fibers and redistribution of focal adhesion proteins is dependent on caveolin-1. This sentence seems not relevant with the context of this part. I cannot understand why the author raised this point here?
  4. Figure 1 seems to be cutoff.
  5. Line147-149: The Senescence Associated Secretory Phenotype (SASP): What do the underlines mean?
  6. Lin223-225: Recently, pre-clinical studies and early clinical trials have highlighted several molecular targets that are effective in the treatment of myeloid malignancies but also may induce, prevent or revert cellular senescence. This sentence is a little bit misleading. How can they induce, prevent or revert senescence, which are supposed to be opposite?
  7. 1 CDK Inhibition and 3.2. Inhibition of MDM2/ MDMX: Potential therapeutic targets in AML treatment through antiproliferation and cell cycle arrest. But the author didn’t state these treatments cause senescence in AML. Because either CDK inhibition or MDM2 inhibition could not necessarily lead to senescence. They can just inhibit cell proliferation or induce apoptosis. If the author wants demonstrate senescence involved in these treatments, relative data or studies should be referenced.

Author Response

  1. Line35: hardly to not detectable.

We agree with the suggestion of reviewer # 2 and have modified the text accordingly.

  1. Line55-57: A hallmark of senescent cells is the exhibition of a senescence-associated secretory phenotype (SASP) mainly consisting of pro-inflammatory cytokines and chemokines, which carries out a complex crosstalk both at the paracrine level and in the microenvironment [20, 21]. Could the author rephrase this sentence? A little bit confusing and incoherent.

We apologize for having been more precise in the first version of our manuscript. The relevant sentence has been re-phrased according to the reviewer’s comments.

  1. Line60-61: The reorganization of the cytoskeleton with an increase in actin stress fibers and redistribution of focal adhesion proteins is dependent on caveolin-1. This sentence seems not relevant with the context of this part. I cannot understand why the author raised this point here?

We thank reviewer #2 for this comment. We do agree that this aspect may not be highly relevant in this context and therefore have deleted the respective part.

  1. Figure 1 seems to be cutoff.

We apologize for this mistake. The layout has been optimized accordingly and delineation of the figure is improved.

  1. Line147-149: The Senescence Associated Secretory Phenotype (SASP): What do the underlines mean?

We apologize for this formatting error. The lines have been removed.

  1. Lin223-225: Recently, pre-clinical studies and early clinical trials have highlighted several molecular targets that are effective in the treatment of myeloid malignancies but also may induce, prevent or revert cellular senescence. This sentence is a little bit misleading. How can they induce, prevent or revert senescence, which are supposed to be opposite?

This is a valid critique of Reviewer #2. We have simplified and rephrased the text.

  1. 1 CDK Inhibition and 3.2. Inhibition of MDM2/ MDMX: Potential therapeutic targets in AML treatment through antiproliferation and cell cycle arrest. But the author didn’t state these treatments cause senescence in AML. Because either CDK inhibition or MDM2 inhibition could not necessarily lead to senescence. They can just inhibit cell proliferation or induce apoptosis. If the author wants demonstrate senescence involved in these treatments, relative data or studies should be referenced.

We agree with the comments of Reviewer#2. Indeed, we aimed to demonstrate sensescence involved in this treatment. Therefore, we have referenced relevant studies in regard to senescence and CDK (Bernard and colleagues) and in regard to senescence and MDM2 (Wiley and colleagues).

Reviewer 3 Report

This review article summarized recent important knowledge of senescence related myeloid malignancies. In my personal opinion, this is a well-organized guide paper for whom may intent to apply senescence in their own study. With a relative minor suggestion from this review. For an academic reader, a table summary would be extremely useful to sort the studies into different drugs/therapy and molecules that correlated to senescence in which myeloid malignancy study. This table may particular focus on the paragraphs of 3.1 to 3.6.

Author Response

We thank Reviewer #3 for this positive assessment of our work and agree on the improvement of the article by adding a table summary. A synoptical table with an excerpt of selected active studies has been included in the revised version of our manuscript.